# Stress among medical students: factor structure of the University Stress Scale among Italian students

Igor Portoghese,[1] Fabio Porru  ,[1,2] Maura Galletta  ,[1] Marcello Campagna,[1] Alex Burdorf  [2]

[1]Department of Medical Sciences and Public Health, University of Cagliari, Cagliari, Sardinia, Italy
[2]Department of Public Health, Erasmus Medical Center, Rotterdam, The Netherlands

**Correspondence to**
Prof. Alex Burdorf;
a.burdorf@erasmusmc.nl

## ABSTRACT

**Objectives** The main purpose of the current study was to investigate the psychometric properties of the Italian version of the University Stress Scale (USS) among Italian medical students.

**Design, setting and participants** A cross-sectional observational study based on data from an online cross-sectional survey from 11 to 23 December 2018. A total of 1858 Italian medical students participated in the study.

**Outcome measures** We measured perceived stress among medical students using the USS, the Effort-Reward Imbalance Student Questionnaire (ERI-SQ) and the Kessler-10 (K10).

**Results** Results showed that a bifactor-Exploratory Structural Equation Modeling solution provided excellent levels of fit to the data. Our results suggest that the modified version of 19 items of the Italian version of the USS does not have a simple unidimensional structure. Overall, an inspection of ancillary indices (omega indices, ECV and percentage of uncontaminated correlations) revealed that these were too low to suggest the use of the USS as a composite measure of university stress. We tested an alternative unidimensional short form (eight items; USS-S) that assessed all the five sources of stress. This version provided a good fit to the data. Evidence of convergent validity of the USS-S was observed by analysing the correlations between the USS and ERI-SQ (ranging from −0.34 to 0.37, all p<0.01). Finally, based on the clinical cut-off recommended on the K10, results from receiver operating characteristic showed that considering the clinical cut-off of the USS is 7.5 and that 59.70% of medical students reported stress levels in the clinical range.

**Conclusion** Finally, our results showed a lack of support for using the USS to measure a general university stress factor, as the general USS factor accounted for little variance in our sample. In this sense, stress scores among Italian students can be better assessed by the use of the USS-S.

## Strengths and limitations of this study

► First study that examined a bifactor solution of the University Stress Scale.
► Short version for the measurement of general stress among Italian medical students.
► Lack of predictive validity.
► Lack of a test–retest reliability.

## INTRODUCTION

Mental health problems are a leading cause of disability and loss of health globally.[1] Mainly, mental health problems and lack of well-being of young individuals have become a public health concern in many countries.[2] In the last decade, many studies showed how the incidence of mental health problems among university students is directly associated with the increase of academic stress.[3–5] High levels of stress and burnout have been described in higher education students worldwide.[6–10] Recently, Deasy *et al* emphasised that the student population is at higher risk of psychological distress when compared with the general population.[11] Academic stress is an important variable that has received researchers' attention in the last 20 years. Rooted in the Lazarus and Folkman's[12] transactional model of stress, academic stress is conceived as a specific relationship between the student and the academic environment that is appraised by the student as taxing or exceeding his or her resources and endangering his or her well-being. Among the main sources of academic stress, there are attending lessons, study overload, respecting deadlines, assessments, financial demands, social pressures and lack of balance between university and private life.[13]

Among university students, special attention was given to medical students.[2] In the last decade, an increasing amount of research has dealt with investigating what makes students' lives more stressful. In fact, medical training has been recognised as a highly stressful experience for medical students.[14 15] A recent review showed that stress prevalence among preclinical students varied between 20.9% and 90%.[16] Several studies showed that the stress of medical training has an impact on the physical and mental health of medical

students.[10 15–19] Furthermore, many academic institutions are implementing interventions aimed at developing coping strategies in students exposed to chronic stressors of medical training.[20]

In this sense, identifying the major stressors within the academic context represents the first step in order to tackle efficiently psychological distress. The most common sources of stress among medical students are: (1) lack of practical skills, (2) feel pressured to get perfect grades, (3) fear of failing, (4) high workload (eg, long hours of study), (5) low free time, (6) quality of the relationships with academic and clinical staff, (7) parental expectations and (8) frequency of examinations.[14 21–23]

Many authors adopted and extended measures of stress, for example, by adapting work-related stress measures to the university context. Recently, Stallman and Hurst developed the University Stress Scale (USS).[24] According to the authors, this is a 'screening measure that captures the cognitive appraisal of demands across the range of environmental stressors experienced by students'. In addition, the authors proposed a (clinical-based) cut-off point as an indicator of the association of levels of stress and mental health problems risk.[24]

According to Stallman and Hurst,[24] the USS was developed with three central purposes: (a) providing a broad measure of different categories of perceived stressors among university students instead of assessing symptoms of stress; (b) not restricting students' perception 'of specific demands and their cognitive appraisal of situations within each category' and (c) providing a reliable self-report measure of perceived stressors among university student population for quickly identifying academic stressors and planning interventions.

Stallman and Hurst defined the USS as composed of six factors: (1) academic, concerning coursework demands, procrastination and study/life balance; (2) equity, concerning discrimination, sexual orientation issues and language/cultural issues; (3) parenting, concerning parenting issues and childcare; (4) relationships, concerning relationships with family, friends and partner and relationship break-down; (5) practical, concerning finances and money problems, housing/accommodation and transport and (6) health, concerning mental and physical health problems.[24]

Concerning the multidimensional structure of the USS, Stallman and Hurst treated the USS as unidimensional in calculating cut-off criteria.[24] In its theorisation, it is not clear if it should be considered as a single unidimensional construct or a collection of different factors. The authors found support for a six factors solution (17 items), but they suggested that 'the strong reliability and validity of the complete measure warrants the inclusion of all items in this instrument'.[24]

In this sense, a bifactor measurement model should represent the better methodological strategy in analysing the structural representation of this measure.[25–31] In fact, in a bifactor measurement model items correlation can be accounted for by (1) a general factor (G-factor)

representing the broad construct of the USS and (2) a set of group factors representing the specific subdomains of the USS.

Furthermore, considering the highly restrictive confirmatory factor analysis (CFA) assumptions and the linked risk of overestimation of factor correlations, we adopted the Exploratory Structural Equation Modelling (ESEM) framework.[32] It is a combination of Exploratory Factor Analysis and CFA, with less restrictive constraints. Specifically, adopting the ESEM framework, items are allowed to load on all the factors.[32] According to many authors, ESEM has been found to represent the factor structure of multidimensional constructs more effectively than CFA.[31 32] ESEM tends to result in a better fit and more accurate estimates, including more realistically estimated factor correlations.[31 32]

Therefore, the purpose of the current study was to (a) investigate the psychometric properties of the Italian version of the USS among Italian medical students and (b) identify clinical cut-offs of the USS to recognise students who are at risk of developing a mental illness.

## METHODS

### Participants and procedure

The target population was medical students in Italy. In Italy, there are 34 Courses of Medicine and Surgery for an estimated population of 154 000 students.[33] Admission to the degree course in Medicine and Surgery is dependent on the results of a national entrance exam. The Italian legislation lays down 6 years of study for the degree course in Medicine and Surgery, including 4 years for compulsory internship/vocational training starting in the third year.

A total of 1858 Italian medical students participated in this cross-sectional study. The sample was recruited through a public announcement at electronic learning platforms for students and university students' associations. The web platforms contained an invitation to join an online survey entitled 'UniCares—Health Promoting University'. The first page of the online protocol described the study's objectives, the time necessary to complete the survey (less than 10 min), the inclusion criteria (being a university student of course in Medicine and Surgery in Italy) and the ethical issues underpinning the study. Participants were informed that their involvement in the study was voluntary and anonymous and that no information that could identify them would be collected. Only individuals who agreed to the study's conditions completed the survey. Furthermore, to ensure anonymity, we did not register the IP address, neither requested any another sensitive data. Finally, the research team did not offer any incentives to increase recruitment neither played any active role in selecting and/or targeting specific subpopulations of students.

The online survey was implemented with LimeSurvey and was available online from 11 to 23 December 2018. The questionnaire investigated sociodemographic

characteristics of the respondents, and three measures of perceived stress.

## Measures

### University Stress Scale

The USS is a 21-item screening measure that captures the cognitive appraisal of demands across the range of environmental stressors experienced by students.[24] Students are asked to rate on a 4-point Likert scale ranging from 'not at all' to 'constantly', the degree to which six stressors (academic, equity, parenting, relationships, practical and health) were a source of stress in the previous month. The English version was first translated into Italian by a bilingual translator, and then this version was translated back into English by another bilingual translator. The two versions were compared, and discrepancies were discussed and changed as needed. Specifically, we removed the parenting subscale (two items) as in Italy the mean age of women at childbirth is 31.8 and 35.3 for men, whereas the mean age of Italian University students is 22.[34] Thus, a version with 19 items was used for this study.

### Students stress

The Effort-Reward Imbalance Student Questionnaire (ERI-SQ) was used for measuring university stress. The Italian version of the ERI-SQ is made up of 12 items that constitute three scales: Effort (2 items; example: 'I have constant time pressure due to a heavy study load'), Rewards (5 items; example: 'I receive the respect I deserve from my supervisors/teachers') and Over-commitment (6 items; example: 'As soon as I get up in the morning I start thinking about study problems').[35] All items are scored on a 4-point rating scale ranging from 1 (strongly disagree) to 4 (strongly agree). The internal consistency for reward and for over-commitment in this sample were adequate, with a McDonald's omega value, respectively. of $\omega=0.69$ and $\omega=0.79$

### Psychological distress

The Kessler 10 (K10) is a brief measure of non-specific psychological distress that discriminates between cases and non-cases of depression and anxiety disorders.[36 37] Its clinical cut-off (K10≥20) in the student population makes it suitable to evaluate the clinical utility of the USS among university students.[8] McDonald's omega was $\omega=0.92$.

## Patient and public involvement

No patient and public involved.

## Statistical analyses

Descriptive statistics were used for sociodemographic information of the population. All models were estimated using the weighted least squares mean and variance adjusted estimator in MPlus V.7.3.[38 39] In the first stage, unidimensional, first-order and second-order CFA, bifactor-CFA, ESEM and bifactor-ESEM were estimated. In first-order CFA, items were allowed to define their a priori factor, no cross-loading was allowed and all factors were allowed to correlate. In second-order CFA, we tested

whether the five first-order factors converge into a second-order factor of university stress. In bifactor-CFA, all items were allowed to define one general factor (G-factor) and a priori specific factors (S-factors). No cross-loading was allowed, and all factors were specified as orthogonal. In first-order ESEM, oblique target rotation was used to allow the free estimation of the main loadings of the items on the a priori factors, while all cross-loadings were estimated but targeted to be as close to 0 as possible. Finally, the bifactor-ESEM solution was estimated using orthogonal target rotation, resulting in the free estimation of items loadings on the G-factor and on one out of S-factors, while all cross-loadings were freely estimated but targeted to be as close to 0 as possible.

Following Morin and colleagues, we began comparing CFA and ESEM solutions.[40 41] In this comparison, as long as the factors remain well-defined by strong target factor loadings, the key issue is related to the factor correlations. Statistical evidence that ESEM tends to provide more exact estimates of true factor correlations suggests that ESEM should be retained whenever the results show a discrepant pattern of factor correlations.[42] Otherwise, the CFA model should be preferred based on parsimony. Then, the second comparison involves contrasting the retained model with its bifactor counterpart (bifactor-CFA or bifactor-ESEM). Here, the key elements favouring a bifactor representation are the observation of a G-factor that is well-defined by strong factor loadings, and the observation of reduced cross-loadings in bifactor-ESEM compared with ESEM. A simplified representation of specified models is shown in figure 1.

For all models, we reported standardised factor loadings (λ, representing the strength of association between each specific item and the underlying factors) and model-based omega coefficients of composite reliability. We relied on the following common goodness-of-fit indices: $\chi^2$, the Comparative Fit Index (CFI), the Tucker-Lewis Index (TLI) and the root mean square error of approximation (RMSEA). In interpreting these fit indices, we considered values >0.90 and 0.95 for the CFI and TLI, respectively, indicating adequate and excellent fit to the

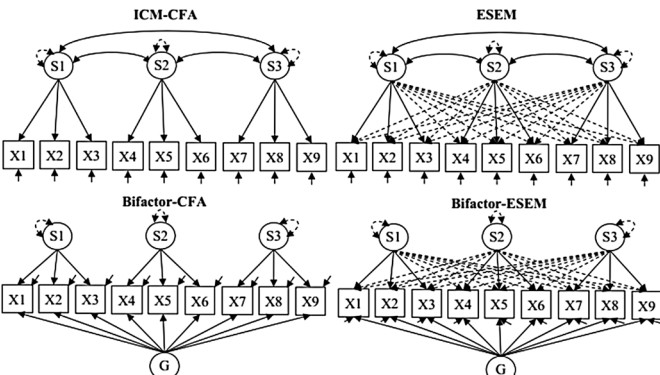

**Figure 1** Simplified representations of specified models. ICM-CFA, independent cluster model of confirmatory factor analysis; ESEM, Exploratory Structural Equation Modelling.

data, whereas values smaller than 0.08 or 0.06 for the RMSEA as acceptable and excellent model fit, and the weighted root mean square residual (WRMR ≤1.00 for acceptable).[32 43]

For model-based reliability, we assessed omega coefficients, including omega for the total score ($\omega$), omega subscale ($\omega_S$), omega hierarchical ($\omega_H$) and omega hierarchical subscale ($\omega_{HS}$).[25 27] $\omega$ and $\omega_S$ provided information about the reliability of the total scale and subscales, respectively. $\omega_H$ provided information about the extent to which composite scale scores were interpretable as a measure of a single common factor. $\omega_{HS}$ indicated the viability of subscales after controlling the variance due to the general factor. Higher $\omega$ and $\omega_S$ scores represent better reliability of the total scale and subscales. Specifically, a $\omega_H$ value higher than 0.80 suggests that the majority of the reliable variance can be attributed to a single general factor.[25] A lower $\omega_{HS}$ value suggests that a higher proportion of the USS score variance can be attributed to the general factor.[25]

Similarly, in order to determine dimensionality, we calculated the explained common variance for the general factor (ECV), the individual item ECV (IECV) and the percentage of uncontaminated correlations (PUC).[27 28] The ECV is the percentage of common variance attributable to the general factor.[25] An ECV ≥0.70 indicates a strong general factor and suggests the common variance is unidimensional.[25] The IECV is the percentage of item common variance attributable to a general factor.[44] The PUC helps in determining whether the bifactor data are unidimensional or multidimensional.[24] PUC>0.70 indicates less bias in structural coefficients and thus indicating that the measure can be treated as unidimensional.[25] According to Rodriguez et al, when ECV is >0.70 and PUC is >0.70, 'relative bias will be slight, and the common variance can be regarded as essentially unidimensional' (Rodriguez et al,[25] p232).

As universally adopted cut-off points for the above indices are not available, we followed Reise and colleagues' suggestion to consider ECV>0.6, PUC>0.7 and $\omega_H$>0.7 in accepting the USS 'unidimensional enough' to warrant the use of a total score.[28] Furthermore, we tested

for convergent validity. Specifically, convergent validity is confirmed inspecting the correlation of USS with another conceptually and theoretically linked measure, the ERI student version. According to Cohen, coefficients≤0.29 were considered low, 0.3 to 0.49 moderate and correlations≥0.5 as high.[45]

Finally, following Stallman and Hurst, procedure, we determined the optimal cut-off point by analysing the receiver operating characteristic (ROC).[24] Specifically, we compared the USS to the clinical cut-off recommended on the K10 for university students (K10≥20). We calculated sensitivities, specificities, positive predictive values (PPV) and negative predictive values (NPV) for the USS. We calculated the areas under the ROC curve (AUC) with 95% CIs. For ROC analyses, we used R.[46]

## RESULTS
### Sociodemographic
The sample for this research consisted of 71.5% females (n=1329). Participants in this study ranged from 19 to 30 years of age (M=23.03, SD=2.17). 16.8% (313) were enrolled in preclinical courses (first and second year) and 83.2% (1545) in clinical courses (years 3–6).

### Psychometric properties of the USS
Table 1 presents the goodness-of-fit indices and information criteria associated with each of the estimated models. The one factor-CFA, the five factor-CFA and the second-order CFA demonstrated inadequate fit. In contrast, the ESEM, the bifactor-CFA and bifactor-ESEM provided excellent representation to the data. However, according to all indices. Generally, comparisons of the models based on the information criteria supported the fit of the B-ESEM model over the corresponding B-CFA.

Bifactor-ESEM solution provided excellent levels of fit to the data (CFI/TLI≥0.90; RMSEA≤0.060). The bifactor-ESEM solution (see table 2) reveals the presence of a G-factor that is relatively well-defined by a majority of items (|λ|=0.17 to 0.67, M=0.38). Further examination of this solution reveals reasonably low cross-loadings,

**Table 1** Goodness-of-fit statistics of the Italian USS

| Model | $X^2$ | df | CFI | TLI | RMSEA (90% CI) | WRMR |
|---|---|---|---|---|---|---|
| One-factor CFA | 2671.97 | 152 | 0.65 | 0.61 | 0.094 (0.091 to 0.097) | 3.27 |
| Five-factor CFA | 1570.48 | 142 | 0.84 | 0.77 | 0.073 (0.070 to 0.076) | 2.46 |
| Second-order CFA | 817.32 | 147 | 0.90 | 0.88 | 0.050 (0.046 to 0.053) | 1.84 |
| One-factor ESEM | 2671.97 | 152 | 0.65 | 0.61 | 0.094 (0.091 to 0.097) | 3.27 |
| Five-factor ESEM | 366.39 | 86 | 0.96 | 0.92 | 0.042 (0.037 to 0.046) | 0.96 |
| Bifactor-CFA | 750.72 | 134 | 0.92 | 0.89 | 0.049 (0.046 to 0.053) | 1.67 |
| Bifactor-ESEM | 192.84 | 72 | 0.98 | 0.96 | 0.30 (0.25 to 0.035) | 0.67 |

n=1858; χ2=Satorra-Bentler scaled chi-square.
CFA, confirmatory factor analysis; CFI, Comparative Fit Index; ESEM, Exploratory Structural Equation Modelling; RMSEA, root mean square error of approximation; TLI, Tucker-Lewis Index; USS, University Stress Scale; WRMR, weighted root mean square residual.

**Table 2** Standardised parameter estimates from the bifactor-ESEM solutions of the USS and factor loadings from CFA solution of the USS-S

| | G (λ) | S1 (λ) | S2 (λ) | S3 (λ) | S4 (λ) | S5 (λ) | USS-S |
|---|---|---|---|---|---|---|---|
| USS1 | 0.23 | **0.80** | *0.00* | *–0.04* | *0.01* | *0.00* | |
| USS2 | 0.42 | **0.13** | *–0.05* | *0.01* | *–0.12* | *0.00* | 0.46 |
| USS3 | 0.36 | **0.46** | *0.06* | *0.06* | *0.11* | *0.07* | |
| USS15 | 0.32 | **0.02** | *0.12* | *0.11* | *0.24* | *–0.01* | |
| USS16 | 0.80 | **0.03** | *–0.04* | *–0.04* | *–0.12* | *–0.23* | 0.66 |
| USS17 | 0.54 | **0.56** | *–0.06* | *–0.01* | *–0.02* | *0.02* | 0.57 |
| USS18 | 0.52 | *0.10* | **0.49** | *–0.05* | *0.08* | *0.03* | 0.54 |
| USS19 | 0.27 | *–0.01* | **0.72** | *0.05* | *–0.15* | *–0.04* | |
| USS20 | 0.18 | *–0.10* | **0.58** | *–0.01* | *0.13* | *0.06* | |
| USS11 | 0.55 | *–0.17* | *–0.10* | **0.03** | *0.13* | *0.18* | 0.55 |
| USS12 | 0.39 | *0.02* | *0.05* | **0.25** | *0.08* | *0.14* | 0.44 |
| USS13 | 0.25 | *0.01* | *–0.04* | **0.95** | *–0.05* | *–0.08* | |
| USS14 | 0.21 | *–0.01* | *0.05* | **0.61** | *0.00* | *0.06* | |
| USS4 | 0.36 | *0.00* | *0.01* | *–0.03* | **0.39** | *0.05* | 0.35 |
| USS5 | 0.32 | *0.01* | *0.06* | *0.02* | **0.64** | *–0.07* | |
| USS6 | 0.18 | *0.05* | *–0.03* | *–0.02* | **0.49** | *–0.04* | |
| USS7 | 0.52 | *0.09* | *–0.03* | *0.05* | *–0.13* | **0.52** | 0.57 |
| USS8 | 0.38 | *0.01* | *–0.02* | *–0.01* | *0.07* | **0.47** | |
| USS21 | 0.30 | *–0.02* | *0.22* | *–0.06* | *–0.10* | **0.23** | |

Bold = target factor loadings.
Italic = non-significant loadings (p > 0.05).
λ, standardised factor loading; CFA, Confirmatory Factor Analysis; ESEM, Exploratory Structural Equation Modelling; G, global factor; S1, specific factor: Academic; S2, specific factor: Equity; S3, specific factor: Relationships; S4, specific factor: Practical; S5, specific factor: Health; USS-S, University Stress Scale Short.

remaining lower than target loadings (|λ|=0.01 to 0.16, M=0.06).

Looking at the S-factors, the Academic (|λ|=0.23 to 0.76, M=0.55), Equity (|λ|=0.21 to 0.69, M=0.48) and Practical (|λ|=0.33 to 0.68, M=0.49) S-factors are relatively fair defined by most items, suggesting that these three S-factors retained specificity over and above that explained by the G-factor. In contrast, the Relationships (|λ|=0.01 to 0.96, M=0.43) and Health (|λ|=0.06 to 0.26, M=0.16) S-factors appear to retain almost no specificity once the variance explained by the G-factor is taken into account, arguing against the added-value of this dimension. Overall, these results support the superiority of the bifactor-ESEM solution, which is retained for further analyses.

For model-based reliability, we assessed omega coefficients. In this study, the ω for the general factor was 0.85, above the recommended threshold of 0.75, indicating that a general factor explained more than 85% of the reliable variance. On the contrary, the $\omega_S$ values were acceptable only for Relationships=0.76 and Equity=0.76, fair for Academic=0.69, and low for Practical=0.61 and Health S-Factors=0.46. The $\omega_H$ value was 0.65, suggesting that 65% of the total USS variance could be attributed to the general factor. Concerning the $\omega_{HS}$ for the five

specific factors were: academic=0.53, relationships=0.44, equity=0.38, practical=0.43 and health=0.02.

Concerning the results about the ECV, IECV and PUC, the common variance associated with the general factor was 0.38, below the 0.70 threshold The ECV for each specific factor for academic, relationships, equity practical and health were 0.16, 0.15, 0.19, 0.11 and 0.01, respectively. The average IECV for the general USS factor was 0.46 and ranged from 0.07 to 0.99. Only four items (USS2, USS11, USS15 and USS16) showed IECV greater than 0.80. Thus, it suggested that, on average, items measured the general factor to a slightly weaker degree than they measured the intended specific factor. Finally, the PUC was 0.82, greater than the suggested 0.70 threshold. However, while higher PUC has been shown decreased the importance of considering the ECV in determining measurement parameter bias, ECV in the present study is not high enough.[25] Overall, these three findings further provided us more information concerning the dimensionality of the USS suggesting that modelling the USS as a unidimensional instrument would lead to substantial measurement parameter bias, and then disqualifying the interpretation of the USS as primarily unidimensional. Therefore, we sought to develop an alternative unidimensional short form of the Italian version of the USS that would consider full coverage of the five sources of stress. In accomplishing this goal, we followed the procedure suggested by Stucky and colleagues[44]: (1) selecting IECV >0.80 and (2) and items with the larger factor loading on the general factor. The items comprising the short form of the USS (USS-S) were: USS2, USS4, USS7, USS11, USS12, USS16, USS17 and USS18. We next assessed the unidimensionality of the USS-S derived from the bifactor-ESEM and IECV analysis. The eight-items unidimensional CFA provided a good fit to the data: CFI=0.94, TLI=0.91; RMSEA=0.068 (95% CI 0.059 to 0.077) and WRMR=1.64. Factor loadings ranged from 0.35 to 0.66, M=0.52) and McDonald's ω was acceptable (0.69).

Evidence of convergent validity was first obtained by analysing the Pearson correlations between the USS-S and the ERI student version. As expected, significant correlations were observed between the USS-S and each of the ERI measures. Specifically, (ranging from −0.34 to 0.37, all p<0.001).

Finally, a ROC curve analysis using K10>=20 as a criterion for possible mental health problems showed that a cut-off of 7.5 on the USS gave an AUC of 0.78 (95% CI 0.76 to 0.80). The sensitivity and specificity of the test were 73.60% (95% CI 71.3% to 75.9%) and 65.20% (95% CI 61.0% to 69.40%), respectively. The PPV of the test was 85.60% (95% CI 83.60% to 87.60%), and NPV of the test was 49.40% (95% CI 45.60% to 53.20%) on psychological distress.

## DISCUSSION

The main purpose of the present study was to investigate the psychometric properties of the Italian version of the USS.

It was designed as a broad assessment of the categories of stressors that are experienced by university students. Stallman and Hurst[24] suggested to use the USS as a general measure of stress, then considering it as a unidimensional measure. In order to gain a better understanding of the strengths of the USS as unidimensional scale, we tested several latent measurement models, including standard CFA, ESEM, bifactor-ESEM and bifactor-CFA.[39 41]

Initially, our results showed that the bifactor-ESEM provided a better fit to the data compared with the other tested models.

Specifically, we found that the USS is composed of a general stress factor and five specific factors, namely Academic, Relationships, Equity, Practical and Health. However, when we considered the ancillary bifactor indices of dimensionality and model-based reliability, results provided us with further information that considering the Italian version of the USS as unidimensional may produce a biased measure of stress. The examination of omega indices, ECV and PUC, suggest that the general factor did not account for enough variance to be considered unidimensional. Specifically, an ECV of 0.38 implies that most of the common variance (62%) is explained by the specific factors. In this sense, Stallman and Hurst[24] never considered this solution and our results suggested that it should be not ignored.

From a theoretical perspective, our findings do not support the existence of a measurable 'overall university stress' construct as suggested by Stallman and Hurst,[24] and the USS is not a reliable measure of general university stress among Italian medical students. In this sense, considering IECV coefficients based on the bifactor-ESEM solution, we derived and tested a short version of this measure (USS-S). This new version showed a clear unidimensional factor structure. This supports the use of the USS-S total score as a measure of the overall stress among medical students.

The items that comprise the shorter version concern study/life balance, procrastination, parental expectations, work, friendships, family relationships, finances and money problems and mental health problems. All items cover the main sources of stress as theorised by Stallman and Hurst.[24]

Furthermore, support for convergent validity of the USS-S was found via the significant correlations with each domain of the ERI-SQ. Although the ERI-SQ has a different theoretical basis, both USS and ERI-SQ are claimed to measure the same general construct.[34 46] Specifically, the USS-S was moderately positively correlated to the Effort subscale. According to the ERI theory, this measure was developed to capture general demands and obligations. The strength of their correlations suggests there is an overlap between these two measures of academic stress. However, further investigation is necessary for understanding the convergent validity of the USS-S.

The clinical cut-off score provides some indication of the level of stress that is likely to impact on student performance, and possibly suggest mental health problems. Given that the majority of students (59.70%) reported stress levels in the clinical range, these findings also support previous research that highlighted the need for preventative interventions to enhance students' realistic appraisals and teach the coping

mechanisms that are necessary to deal with stressful situations.[8] This will help prevent the challenges that are associated with university study from becoming distressing, thereby placing students at risk for the development of psychological disorders.

In general, outcomes from our study suggest that the Italian version of the USS is a multidimensional measure characterised by a global university stress factor, and by specific factors. Both G-factor and S-factors should be considered in measuring stress among university students when measured by the USS. However, researchers interested in adopting this measure are strongly encouraged to verify its dimensionality during preliminary analyses.

The present study has several limitations. First, we collected data from a convenience sample that reduced the generalisability of our results. That implies that the principles of probability sampling were not followed and our results could be biased. However, considering the aim of the present study to assess the psychometric properties of the USS among Italian medical students, this sample was considered appropriate. Second, as we derived a shorter version of the USS, we strongly encourage future research to investigate psychometric properties of this version among other cultural groups. Third, this study lacks a test–retest reliability, which could test the consistency of the measures over time. Fourth, we did not investigate measurement invariance across gender and different languages. Future research should explore that in order to increase knowledge of university stress from a cross-cultural perspective. Finally, we did not investigate the predictive validity of the Italian version of the USS-S. Future research should explore, for example, the ability of the USS to predict psychological well-being among students.[26]

**Contributors** IP, FP, MG, MC and AB contributed to the conception and design of the study. IP, FP and MG and contributed to the development procedure of the Italian version of USS, including forward translation and back translation review. FP contributed to the acquisition of data. IP analysed the data. FP wrote the first draft of the manuscript. IP and MG supervised the analysis. IP, FP, MG, AB and MC revised the manuscript. All authors read and approved the final version of the manuscript.

**Funding** The authors have not declared a specific grant for this research from any funding agency in the public, commercial or not-for-profit sectors.

**Competing interests** None declared.

**Patient consent for publication** Not required.

**Ethics approval** This study was developed in accordance with the ethical standards of the institutional and/or national research committee and with the 1964 Helsinki declaration and its later amendments.

**Provenance and peer review** Not commissioned; externally peer reviewed.

**Data availability statement** Data are available upon reasonable request. Raw data pertaining to analyses performed in this study are available from the authors upon reasonable request.

**ORCID iDs**
Fabio Porru http://orcid.org/0000-0001-9202-6168
Maura Galletta http://orcid.org/0000-0002-0124-4248
Alex Burdorf http://orcid.org/0000-0003-3129-2862

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
