## [Reviewer comments · BMJ Open]

ARTICLE DETAILS

TITLE (PROVISIONAL)	Stress Among Medical Students: Factor Structure of the University Stress Scale Among Italian Students
AUTHORS	Portoghese, Igor; Porru, Fabio; Galletta, Maura; Campagna, Marcello; Burdorf, Alex

VERSION 1 – REVIEW

REVIEWER	Maciej Walkiewicz Faculty of Psychology Medical University of Gdansk, Poland
REVIEW RETURNED	21-Nov-2019

GENERAL COMMENTS	Thank you for referring to me manuscript bmjopen-2019-035255 entitled „Stress Among Medical Students: bifactor measurement model of the University Stress Scale". This study addresses an important topic. Nevertheless, there are some concerns that should be diligently addressed before considering this manuscript for publication in the BMJ Open. First, in the Methods section, the Authors should clearly explain who exactly took part in the survey e.g. medical students of which years - clinical or preclinical. More information is needed about the study group (sociodemographic data) and the study procedure. Second, for an international reader, it can be helpful to add information on how the recruitment process looks for medical schools in Italy, and how the education system looks in general. How many schools are there and how many students. What percentage of medical students is N=1858. Third, for me, it was not clear why the Authors want to check the USS, whether only with medical students and not, e.g., nursing, emergency medicine, or healthcare education in general. Fourth, why the ERI-SQ and K10 tools were used for validation and not any others. Fifth, If I were the Authors, I would consider updating the literature, especially references: 8,10,11,23,32,33. Sixth, I agree with the opinion that the tool would be useful in medical education, seeking the risk of developing a mental illness to cover support. Therefore I have a question - how the Authors would see the application of the tool. Do the authors have any idea how to conduct such an examination
---

	at a medical university and propose interventions for students, e.g. "reported stress levels in the clinical range"?
--	--

REVIEWER	Alex Garn Louisiana State University
REVIEW RETURNED	04-Dec-2019

GENERAL COMMENTS	The purpose of this study was to investigate the psychometric properties of the University Stress Scale (USS) in a sample of Italian medical students. Specifically, the focus was on examining a bifactor measurement model using the exploratory structural equation model (ESEM) analytical framework. Below I provide comments and suggestions on the manuscript.  1. Please revise the first sentence of the introduction by adding "Mental health disorders are a leading cause ..." or another descriptor that you prefer. Mental health is typically framed as positive functioning (see World Health Organization definition) so I this sentence needs clarification. 2. Line 63: Please revise the sentence that ends with [12]. Stress levels is a pretty generic term and it is unclear what having a certain % stress level means. Please add context. 3. Please revise the sentence starting on line 70 that outlines the sources of stress among medical students. The initial wording is awkward. Change to something like "The most common sources of stress among medical students are ..." 4. In the second to last paragraph of the introduction where you discuss ESEM, I think it is important to add context to the concept of stress in general and more specifically the USS. Why would you expect there to be cross-loadings with the USS scale? Is there evidence of cross-loadings in the USS development work by Stallman and Hurst? 5. Is it possible to provide more description about the sample of students? For example, what was the age (mean and standard deviation) of the these students? What is the breakdown of males and females? How many different medical schools did they come from? How many different regions of Italy? Etc. 6. Line 130: Please make it clear that the parenting subscale consisted of only two items so it is clear that you removed this entire dimension. 7. It is a little unclear in your measures section why you report coefficient alpha for the ERI-SQ and K10 rather than coefficient omega, which is a model-based internal consistency measure and superior to coefficient alpha. Furthermore, you do not report the internal consistency estimate for the Effort subscale of the ERI-SQ. 8. Typically, when ordered categorical scales with less than 5 options are used in CFA/ESEM, diagonally weighted least squares (WLSMV) estimation procedures produce less biased estimates compared to maximum likelihood / robust maximum likelihood procedures which assume continuous indicators. Please provide justification for using robust ML or rerun you analyses with WLSMV. 9. Many bifactor experts that you cite in this manuscript (e.g., 22-24) highlight the importance of supplementing model comparisons with factor strength analyses that provide greater context on relations between the general factor and specific factors. Specifically, adding estimates for the Explained Common Variance (EVC) and Percent Uncontaminated Correlations (PUC) are two additional estimates that I encourage you to add to this manuscript. 10. Similarly, in a bifactor model it is recommended to report the coefficient omega hierarchical for the general factor and coefficient
--

	omega hierarchical subscale internal consistency estimates rather than coefficient omega. With coefficient omega, each specific factor (e.g., academic stress) is infused with true score variance from the general factor and specific factor of interest. Coefficient omega hierarchical for the general factor and coefficient omega hierarchical subscale for each specific factor decompose these two sets of true score variance so you have a more accurate picture of internal consistency of the subscales scores after controlling for the general factor. See 22 from your reference list. 11. Please add a correlation matrix from the five factor CFA so readers have a better understanding of the multidimensionality relations of the USS specific factors. 12. The AIC, BIC, and ABIC values are not provided in Table 1 yet there are columns for these values. Thus, either delete these columns or add the values and a description in the statistical analyses section. 13. Is there a reason that you do not provide the cross-loadings in Table 2 for the ESEM findings? Without providing these values or included a note about why they are not included, readers might assume these is a bifactor CFA. 14. I think the manuscript would be strengthened by adding more detail on the convergent validity findings. 15. The overall pattern of factor loadings for the G factor and specific factors are low. For example, all standardized factor loadings for the G factor except 1 are below .60. Furthermore, there are no specific factors with a set of factor loadings above .50. These parameter results suggest that the USS may be an unreliable measure of stress for this sample. This is currently not addressed in the Discussion. 16. I think that the discussion can be enhanced after incorporating in the EVC, PUC, coefficient omega hierarchical, and coefficient omega hierarchical subscale values.
--	---

REVIEWER	Kaj Sparle Christensen Institute of Public Health Aarhus University Denmark
REVIEW RETURNED	06-Mar-2020

GENERAL COMMENTS	Interesting paper. Abstract: Please provide information on data sampling period. Please provide data on sex and age of participants. Limitations of this study: I suggest writing 1) no assessment of invariance across sex, age and languages, 2) no assessment of test-retest reliability, and 3) no assessment of predictive validity Introduction: L54 'many studies showed', please provide more than 1 reference. Methods: Please provide information on data sampling period and demographics. L141-142: Do you mean Chrobachs alpha? Value of alpha for effort subscale? Value of alpha for ERI-SQ? Results: L232: Please specify: 'ROC curve analyses using K10 ≥ 20 as criterion for possible mental health problems, provided...'. Please provide 95% CI intervals for sensitivity, specificity, PPV and NPV. Discussion: Please discuss whether your sampling procedure is
---

	likely to cause oversampling of individuals with mental health problems. Please include the Strobe checklist.
--	--

VERSION 1 – AUTHOR RESPONSE

R1.1. First, in the Methods section, the Authors should clearly explain who exactly took part in the survey e.g. medical students of which years - clinical or preclinical. More information is needed about the study group (sociodemographic data) and the study procedure.

◇ We added more details about the study population and the procedure we followed in collecting data.

R1.2. Second, for an international reader, it can be helpful to add information on how the recruitment process looks for medical schools in Italy, and how the education system looks in general. How many schools are there and how many students. What percentage of medical students is N=1858.

◇ Concerning the Italian context, we briefly described the admission process, the number of Courses and the organization of the Course of Medicine and Surgery in Italy.

R1.3. Third, for me, it was not clear why the Authors want to check the USS, whether only with medical students and not, e.g., nursing, emergency medicine, or healthcare education in general.

◇ As we described on lines 60-65, "Among University students, special attention was given to medical students, [2]. In the last decade, an increasing amount of research has dealt with investigating what makes students' lives more stressful. In fact, medical training has been recognized as a highly stressful experience for medical students, [10,11]. A recent review showed that stress levels among preclinical students varied between 20.9 and 90%, [12]."

The rationale behind our study was to test the psychometric properties the Italian version of the USS in a recognized "stressed" subpopulation of university students. For example, Dyrbye and colleagues (2010) developed the Medical Student Well-Being Index. Al-Dubai and colleagues (2012) tested the psychometric properties of the Malay Version of the Perceived Stress Scale among Malaysian Medical Students. Zhang and colleagues (2018) tested the psychometric properties of the Chinese version of the Communication Skills Attitude Scale among medical students. We hope that helps in clarifying this point.

R1.4. Fourth, why the ERI-SQ and K10 tools were used for validation and not any others.

◇ Actually, the number of measures aimed to assess stress among university (and medical) students is very limited. Furthermore, the ERI-SQ and K10 are the only measures available in Italian. Both are questionnaire developed for this purpose and we opted for those because the theoretical rationale behind their development.

R1.5. Fifth, If I were the Authors, I would consider updating the literature, especially references: 8,10,11,23,32,33.

◇ We updated the first 4 references. The last two refer to the original validation of the K-10.

R1.6 Sixth, I agree with the opinion that the tool would be useful in medical education, seeking the risk of developing a mental illness to cover support. Therefore I have a question - how the Authors would see the application of the tool. Do the authors have any idea how to conduct such an examination at a medical university and propose interventions for students, e.g. "reported stress levels in the clinical range"?

◇ The main purpose of our study was to test the psychometric properties of the USS among Italian medical students. In this sense, the reviewer's question is very important but it is beyond the scope of

our manuscript. However, our main contribution is related to the identification of a valid measure of stress among university students. Universities should consider to adopt valid and reliable measures. Furthermore, as stress is not a pathology but could expose students to mental illness, it is fundamental to use measure that investigate more different source of stress, such as academic, equity, parenting, relationships, etc. Furthermore, the short version we proposed may help universities in monitoring a general source of stress among medical students.

Concerning the clinical range of our sample, we wrote that “it is likely to impact upon student performance, and possibly suggest mental health problems”. This does not mean that students in high stress show any kind of mental health problem.

R2.1. Please revise the first sentence of the introduction by adding “Mental health disorders are a leading cause ...” or another descriptor that you prefer. Mental health is typically framed as positive functioning (see World Health Organization definition) so I this sentence needs clarification.

◇ You are right. Thank you for your suggestion. We fixed it.

R2.2. Line 63: Please revise the sentence that ends with [12]. Stress levels is a pretty generic term and it is unclear what having a certain % stress level means. Please add context.

◇ We changed the work “levels” with prevalence, as suggested in the review we referred to.

R2.3 Please revise the sentence starting on line 70 that outlines the sources of stress among medical students. The initial wording is awkward. Change to something like “The most common sources of stress among medical students are ...”

◇ We fixed it changing this sentence with “The most common sources of stress among medical students are:”.

R2.4. In the second to last paragraph of the introduction where you discuss ESEM, I think it is important to add context to the concept of stress in general and more specifically the USS.

◇ On lines 70-77 we expanded the definition of academic stress. On lines 102-108 we briefly described the ratio behind the development of the USS.

R2.4. Why would you expect there to be cross-loadings with the USS scale? Is there evidence of cross-loadings in the USS development work by Stallman and Hurst?

◇ The main problem about the USS is that in its original theorization, Stallman and Hurst treated the USS as unidimensional in calculating cut-off criteria. That requires a bifactor structure. In this sense, concerning the multidimensional structure of the USS, we investigated that by adopting a bifactor perspective. That is the most important point. As we described in our manuscript “a bifactor measurement model should represent the better methodological strategy in analyzing the structural representation of this measure”. In this sense, using the USS as unidimensional without providing evidence of that may produce a biased measure of stress.

R2.5. Is it possible to provide more description about the sample of students?

For example, what was the age (mean and standard deviation) of these students? What is the breakdown of males and females? How many different medical schools did they come from? How many different regions of Italy? Etc.

◇ We described demographics at lines 150-153.

R2.6. Line 130: Please make it clear that the parenting subscale consisted of only two items so it is clear that you removed this entire dimension.

◇ We changed this sentence as follows: “Specifically, we removed from this study the parenting

subscale (two items) as in Italy the mean age of women at childbirth is 31.8 and 35.3 for men...”.

R2.7. It is a little unclear in your measures section why you report coefficient alpha for the ERI-SQ and K10 rather than coefficient omega, which is a model-based internal consistency measure and superior to coefficient alpha.

◇ Thank you very much for your valuable suggestion. We updated this section by referring to McDonald's omega.

R2.7 Furthermore, you do not report the internal consistency estimate for the Effort subscale of the ERI-SQ.

◇ This subscale is made up of just 2 items and then it is not possible to calculate any internal consistency.

R2. 8. Typically, when ordered categorical scales with less than 5 options are used in CFA/ESEM, diagonally weighted least squares (WLSMV) estimation procedures produce less biased estimates compared to maximum likelihood / robust maximum likelihood procedures which assume continuous indicators. Please provide justification for using robust ML or rerun you analyses with WLSMV.

◇ Our analyses were based on the robust maximum likelihood (MLR) estimator providing standard errors and fit indexes that are robust to the Likert nature of the items and violations of normality assumptions (Morin, Arens, and Marsh, 2015). MLR has its own strengths – e.g., generally less biased standard error estimates of factor loadings and structural coefficients, and accurate and precise structural coefficient estimates in the conditions of symmetric data. MLR does not require a large sample to produce stable structural coefficient estimates and standard error estimates of factor loadings and structural coefficients, but may need a quite large sample (e.g., $N = 1,000$ or more) to control for Type I errors of testing overall model fit, despite the existence of moderate underestimation in factor loading estimates.

However, we recognize the important point highlighted by the review and strongly agree with it. In this sense, we reperformed all the analyses by adopting the categorical WLSMV estimator. As this estimator does not offer AIC and BIC, we removed these indices from our manuscript.

R2.9. Many bifactor experts that you cite in this manuscript (e.g., 22-24) highlight the importance of supplementing model comparisons with factor strength analyses that provide greater context on relations between the general factor and specific factors. Specifically, adding estimates for the Explained Common Variance (ECV) and Percent Uncontaminated Correlations (PUC) are two additional estimates that I encourage you to add to this manuscript.

◇ We agree with the reviewer's suggestion. It is not common finding reviewers with expertise in bifactor methodology. We added both values for ECV and PUC, and these helped us in understanding the dimensionality of the USS. Furthermore, we considered the individual item explained common variance (IECV). We described ECV, IECV, and PUC on line 230-245. Thank you very much.

R2.10. Similarly, in a bifactor model it is recommended to report the coefficient omega hierarchical for the general factor and coefficient omega hierarchical subscale internal consistency estimates rather than coefficient omega. With coefficient omega, each specific factor (e.g., academic stress) is infused with true score variance from the general factor and specific factor of interest. Coefficient omega hierarchical for the general factor and coefficient omega hierarchical subscale for each specific factor decompose these two sets of true score variance so you have a more accurate picture of internal consistency of the subscales scores after controlling for the general factor. See 22 from your reference list.

◇ That is another important point highlighted. We appreciated the reviewer's suggestion and then we added omega coefficients. We described these coefficients on lines 219-229. Specifically, for model-

based reliability, we assessed omega coefficients, including omega for the total score (ω), omega subscale (ω_S), omega hierarchical (ω_H), and omega hierarchical subscale (ω_{HS}).

R2.11. Please add a correlation matrix from the five factor CFA so readers have a better understanding of the multidimensionality relations of the USS specific factors.

◇ As our results do not support the five factor CFA, it is not a valuable information and, in our opinion, it might confuse the readers.

R2.12. The AIC, BIC, and ABIC values are not provided in Table 1 yet there are columns for these values. Thus, either delete these columns or add the values and a description in the statistical analyses section.

◇ As we reperformed all the analyses by adopting the categorical WLSMV estimator, AIC and BIC are not available. We removed these indices from our manuscript.

R2.13. Is there a reason that you do not provide the cross-loadings in Table 2 for the ESEM findings? Without providing these values or included a note about why they are not included, readers might assume these is a bifactor CFA.

◇ The reviewer is right. We updated the table as requested.

R2.14. I think the manuscript would be strengthened by adding more detail on the convergent validity findings.

◇ We added more details about the convergent validity of the USS short version.

R2.15. The overall pattern of factor loadings for the G factor and specific factors are low. For example, all standardized factor loadings for the G factor except 1 are below .60. Furthermore, there are no specific factors with a set of factor loadings above .50. These parameter results suggest that the USS may be an unreliable measure of stress for this sample. This is currently not addressed in the Discussion.

◇ Actually, the only suggested cut-off for considering factor loading in bifactor analysis is the Thurstone's classical criterion for "salience" (1947) as standardized factor loading being $>.30$. However, factor loadings are not a reliable index in bifactor methodology. As we showed, other important coefficients should be considered and the decision to consider the USS as a not reliable measure of stress was based on omegas, ECV, and PUC. We addressed that in the discussion.

R2.16. I think that the discussion can be enhanced after incorporating in the EVC, PUC, coefficient omega hierarchical, and coefficient omega hierarchical subscale values.

◇ As we have followed all the suggestions of the reviewers, in the light of the results obtained with the new analyses, we have updated the discussion.

Reviewer 3.

R3.1. Abstract: Please provide information on data sampling period.

◇ We added this information in the abstract.

R3.2. Please provide data on sex and age of participants.

◇ We described demographics of our sample in the text.

R3.3. Limitations of this study: I suggest writing 1) no assessment of invariance across sex, age and languages, 2) no assessment of test-retest reliability, and 3) no assessment of predictive validity

◇ Thank you for this important suggestion. We added and discussed these limitations.

R3.4. Introduction: L54 'many studies showed', please provide more than 1 reference.

◇ We added other references in support of our statement.

R3.5. Methods: Please provide information on data sampling period and demographics.

◇ We added these details in the sections entitled "Participants and Procedure" and "Demographics" (lines 139-153).

R3.6. L141-142: Do you mean Cronbach's alpha? Value of alpha for effort subscale? Value of alpha for ERI-SQ?

◇ Effort sub-scale was made up of two items and that is the reason why we did not report alpha.

Furthermore, we did not consider the ERI-SQ as a global measure of stress but we investigated the relationship of each ERI sub-dimension with the USS.

R3.7. Results: L232: Please specify: 'ROC curve analyses using K10 ≥ 20 as criterion for possible mental health problems, provided...'. Please provide 95% CI intervals for sensitivity, specificity, PPV and NPV.

◇ Thank you for your important suggestion in clarifying the criterion of our ROC analysis. Furthermore, we added 95% CI intervals for sensitivity, specificity, PPV and NPV.

R3.8. Discussion: Please discuss whether your sampling procedure is likely to cause oversampling of individuals with mental health problems.

◇ We agree with the reviewer's point. We really appreciated this suggestion. Sampling errors are an important issue when researchers' aim to investigate a phenomenon, such as the relationship between two or more variables (for example, the effect of study overload and support from university staff on student perceived stress). In fact, collecting data from internet may lead to biased data if researchers do not adopt a probability sampling paradigm. But our study did not investigate that. The only way to reduce this bias is to know the prevalence and distribution of mental disorders among Italian Medical students and then plan a sampling procedure that would consider that. It would be a great study but the aim of our study was very limited to a pure psychometric question.

Concerning our study, we are not able to say that our sampling procedure generated data with an oversampling of students with mental health problems. In fact, our data are in line with previous studies that have reported high percentage of students with high stress levels (in our study 59.70%). We are aware that self-selection may expose researchers to collect data from specific cluster of respondents. But that is the same problem as in a paper survey.

R3.9 Please include the Strobe checklist.

◇ We included it as supplementary material.

VERSION 2 – REVIEW

REVIEWER	Maciej Walkiewicz Medical University of Gdansk, Poland Chair of Psychology
REVIEW RETURNED	18-Jun-2020

GENERAL COMMENTS	Thank you for the answers. I think that the Authors referred to the comments and recommendations correcting fragments of the manuscript or answering in correspondence. I hope that research will contribute to the improvement of medical education in Italy.
--

REVIEWER	Kaj Sparle Christensen Institute of Public Health, Aarhus University, Aarhus, Denmark
-----------------	--

REVIEW RETURNED	17-Jun-2020
GENERAL COMMENTS	The authors have responded satisfactory and corrected the paper based on my comments.